Postembryonic development of the tracheal system of beetles in the context of aptery and adaptations towards an arid environment

Raś Marcin mras@miiz.waw.pl 1
Wipfler Benjamin 2
Dannenfeld Tim 2
Iwan Dariusz 1
1 Zoological Museum, Museum and Institute of Zoology, Polish Academy of Sciences , Warsaw , Poland
2 Zoologisches Forschungsmuseum Alexander Koenig, Leibniz-Institut zur Analyse des Biodiversitätswandels , Bonn , Germany
Gillespie Joseph
Electronic publication date: 2022 Jul 13
Publication date: 2022
Volume: 10
Electronic Location ID: e13378
Received 2021 Oct 4; Accepted 2022 Apr 13
Copyright: ©2022 Raś et al.
Copyright year: 2022
Copyright holder: Raś et al.
License: This is an open access article distributed under the terms of the Creative Commons Attribution License, which permits unrestricted use, distribution, reproduction and adaptation in any medium and for any purpose provided that it is properly attributed. For attribution, the original author(s), title, publication source (PeerJ) and either DOI or URL of the article must be cited.
License URL: https://creativecommons.org/licenses/by/4.0/

Keywords: Respiration, Wing reduction, Desert beetles, Morphometrics, Tracheal system

Funding: Preludium 12 Project from the National Science Center, Poland Number 2016/23/N/NZ8/02815 Deutsche Forschungsgemeinschaft WI 4324/4-1 This research was funded by the Preludium 12 Project (Number 2016/23/N/NZ8/02815) from the National Science Centre, Poland and the Deutsche Forschungsgemeinschaft (WI 4324/4-1). The funders had no role in study design, data collection and analysis, decision to publish, or preparation of the manuscript.

==============================
The tracheal system comprises one of the major adaptations of insects towards a terrestrial lifestyle. Many aspects such as the modifications towards wing reduction or a life in an arid climate are still poorly understood. To address these issues, we performed the first three-dimensional morphometric analyses of the tracheal system of a wingless insect, the desert beetle Gonopus tibialis and compared it with a flying beetle (Tenebrio molitor). Our results clearly show that the reduction of the flight apparatus has severe consequences for the tracheal system. This includes the reduction of the tracheal density, the relative volume of the trachea, the volume of the respective spiracles and the complete loss of individual tracheae. At the same time, the reduction of wings in the desert beetle allows modifications of the tracheal system that would be impossible in an animal with a functional flight apparatus such as the formation of a subelytral cavity as a part of the tracheal system, the strong elongation of the digestive tract including its tracheal system or the respiration through a single spiracle. Finally, we addressed when these modifications of the tracheal system take place during the development of the studied beetles. We can clearly show that they develop during pupation while the larvae of both species are almost identical in their tracheal system and body shape.

Introduction

Insects comprise the most successful lineage of organisms on this planet in terms of described species (Stork et al., 2015). One of the major driving forces for this extreme evolutionary success is the ability to fly, which the group evolved around 420 million years ago (Misof et al., 2014) and thus much earlier than in any other group of animals. However, this mode of locomotion is also extremely demanding as it requires a massive flight musculature, a challenging neural control of the 3D environment and enormous amounts of energy. As a consequence, wing reduction is a commonly observed adaptation in almost any lineage of insects (Roff, 1990). These reductions are promoted by factors such as high costs of dispersion or stable environmental conditions which do not require long distance migration (Roff, 1990) and can range from the shortening of the wings, over strong reductions (brachyptery), to a complete loss of all elements of the flight apparatus (aptery) (Heppner, 1991). A group particularly prone to winglessness are darkling beetles (Tenebrionidae) that inhabit arid regions. In the Namib Desert for example, it was shown that 98% of the tenebrionid beetles are wingless (Koch, 1961).

Flight with its massive musculature is also extremely demanding in terms of oxygen and thus energy supply. Insects respire with tracheae, pipe-like structures that form a complex three-dimensional system throughout the entire animal (Dittrich & Wipfler, 2021). They are connected to the outside via spiracles, slit-like openings in the lateral body wall. Each fibril of an insect’s flight muscles is surrounded by huge mitochondria surrounded by tracheoles (Edwards & Ruska, 1955; Wigglesworth & Lee, 1982). As a result, flying insects have a 30 times greater mass-specific oxygen consumption rate than terrestrially locomoting ones (Harrison & Roberts, 2000). It is thus hypothesized that the two thoracic body segments associated with flight (the meso- and metathorax) are much better supported by the tracheal system in winged insects than in apterous ones. However, no study ever addressed this correlation in a comparative approach or the nature and expression of these differences. Additionally, several other questions in the context of the tracheal system and wing reduction are unresolved. Does the reduction of the flight apparatus and the subsequent relief of the tracheal system allow specific adaptations of the respiratory system that would be impossible in a winged insect? When do these modifications of the tracheal system occur during the development as wings are only present in the adult insects? This is particularly interesting in the insects with a complete metamorphosis (Holometabola). In this group, the larvae can have a completely different habitus, lifestyle, physiology and diet as the adults, as for example the caterpillar and the butterfly. This is enabled by a pupal stage during which almost all tissues, systems, and organs of the body are reconstructed and reorganized (Nicholson, Ross & Mayhew, 2014; Ferns & Jervis, 2016).

The tracheal system was traditionally poorly studied as it needed laborious preparation and it was difficult to transfer the extremely complex three-dimensional structure into a 2D drawing (e.g., Wigglesworth, 1954; Perenau et al., 2007). With the introduction of new techniques like micro-computed tomography, the study of the tracheal system experienced a renaissance which also affected the study of the development of the tracheal system during metamorphosis. Examples include the painted lady butterfly Vanessa cardui (Lowe et al., 2013), the blowfly Calliphora vicina (Hall, Simonsen & Martín-Vega, 2017) or the solitary bee Megachile rotundata (Helm et al., 2018). Additionally, the question between the scaling of the tracheal system and the body volume was addressed (Kaiser et al., 2007). These methods also allow the analysis of the tracheal system in live insects and showed for example that parts of the tracheal system undergo contraction during respiration (Socha et al., 2008; Waters et al., 2013) or how the tracheal systems adjusts to hypoxia (Greenlee et al., 2013).

The present study aims on addressing the above stated questions with sophisticated three-dimensional approaches such as virtual 3D models based on µ-computed tomography, 3D morphometrics, Gaussian mixture models and principal component analyses. Specifically, the following hypotheses will be tested: (I) the pterothoracic (i.e., the segments that hold the wings) tracheal system of the studied unwinged species is significantly reduced in comparison to the winged one. (II) The reduction of wings in the studied desert beetle creates capacities in the tracheal system for specific adaptations towards this arid environment that are also found in other beetles of arid environment (III) the observed specializations develop during the pupal stage in the holometabolous beetles. As model system to address these hypotheses we selected two darkling beetles (Tenebrionidae), the apterous Gonopus tibialis (Fabricius, 1798) that inhabits the dry areas of South Africa (Kamiński et al., 2018) and the winged Tenebrio molitor Linnaeus, 1758, which is a model cosmopolitan species considered as a warehouse pest. For the latter species, we rely on the raw data provided by Raś, Iwan & Kamiński (2018).

Materials & Methods

Animals

The present study is based on specimens of Gonopus tibialis (Fabricius, 1798) and Tenebrio molitor Linnaeus, 1758. The larval and pupal specimens of G. tibialis were bred from identified adults collected in the Kuzikus Wildlife Reserve, Namibia (−23.229539, 18.401057; 9.10.2012; coll. D. Schimrosczyk & J. Reinhard, permit for research and collecting nr 1738/2012 granted by Ministry of Environment and Tourism of Namibia). For G. tibialis complete tracheal models for nine larvae, five pupae and nine imagines were obtained. The analyses performed for T. molitor are based on the raw data from Raś, Iwan & Kamiński (2018) and includes ten adults, nine pupae and ten larvae.

3D model generation

μ-CT scanning and subsequent model reconstruction follows the protocol of (Iwan, Kamiński & Raś, 2015). µ-Ct scanning was performed with a Bruker SkyScan 1172 with the following parameters: voltage: 40 kV, current: 250 mA, exposure time: 589 ms–1,178 ms, rotation steps: 0.2° over 180°, frame averaging: 3, random movement: off/5. The resolution of the obtained scans ranged from 4.3 to 12.3 µm pixel size depending on the size of the specimen. The reconstructions were performed with NRecon (ver. 1.7.1.0). The segmentation process followed by tissues and tracheal system volume measurements were done with CTAn 1.18.4.0+ (Bruker). 3D Visualization was performed in Blender 2.8 (https://www.blender.org/).

Definitions and terminology

In order to compare the tracheal system of the different body parts, we defined three tracheal sections based on the tagmata and body segments of the beetles: the cephalo-prothoracic section which comprises all tracheal branches situated in head and prothorax, the pterothoracic section which consists of the trachea situated in the meso- and metathorax, the meso- and metathoracic spiracles, the legs, wings and elytra and the abdominal section with all tracheae and spiracles in the abdomen. The nomenclature of the tracheal structures follows Raś, Iwan & Kamiński (2018) and is provided with some modifications in Raś (2020, Table 1).

Morphometrics and statistical analyses

Morphometric measurements of the examined structures were performed in Blender 2.8 (https://www.blender.org/). R (ver. 3.5.1) (R Core Team, 2017) was used for statistical analysis and presentation of results with the geomorph (ver. 3.0.7), ggplot2 (ver. 3.0.0), mclust (ver. 5.4.1) and smatr (ver. 3.4–8) packages. Total body volume consists of body tissues without tracheal system (spiracles and trachea) and subelytral cavity volumes. The volume of individual spiracles was measured with Blender using the mesh volume measurement tool. The coordinates of the spiracles were obtained from 3D models and analyzed using the geomorph package for R (Adams et al., 2019). A Generalized Procrustes Analysis was used to analyze data and obtain shape information. Averaged models of individual developmental stages of both G. tibialis and T. molitor were presented with thin-plate spline grids. The analysis by Gaussian mixed model (mclust package) (Scrucca et al., 2016) were based on a set of coordinates (x, y, z) which describe the vertices of the 3D model of the tracheal system. The concentration of the tracheal system of the studied beetles (adult stages, n = 3 for each species) was determined by using density function in the mclust package. Growth rate (allometry) analysis of the tracheal system was performed with the smatr package, and was determined in relation to the body tissue volume. The Wilcox Rank Sum Test was used to compare volumes of multiple spiracles between each other. It was performed in R. The Kruskal-Wallis rank sum test was used to compare volumes of body tissue between development stages of both studied species. The diameter of the trachea that run posteriorly from the mesothoracic spiracle was measured at the maximum cross section in the middle between the meso- and metathoracic spiracles for 5 specimens of G. tibialis and 10 specimens of T. molitor.

Digestive system

The digestive system was studied for additional five adult specimens of each species by scanning dried specimens. The applied protocol follows Shively & Miller (2009). The volumes of the respective digestive systems were measured.

All raw data used for the analyses and complete results of all measures are available on Harvard Dataverse (Raś, 2020).

Results

Morphology of the tracheal system (Figs. 1 and 2)

All studied stages of G. tibialis and T. molitor have a holopneustic tracheal system which comprises ten pairs of spiracles (meso- and metathorax and abdominal segments I–VIII) (Fig. 1). Figure 1 also shows the three defined sections of the tracheal systems of all studied species and stages.

Figure 1 Three-dimensional reconstructions of the tracheal systems of the larva (A, B), pupa (C, D) and image (E, F) of Gonopus tibialis (A, C, E) and Tenebrio molitor (B, D, E) in dorsal view.

Cephalic-prothoracic section in yellow, pterothoracic section in red and abdominal section in blue. Thoracic (ts1 & ts2) and abdominal (as1-as8) spiracles illustrated in the right body half.

The larva

The tracheal system of the larva of G. tibialis is shown in Figs. 1A & 2A.

Figure 2 Tracheal system of the anterior body half (reaching towards abdominal spiracle 3) of the larva (A), pupa (B) and the imago (C) of Gonopus tibialis in lateral view.

Dorsal longitudinal trunk in purple, ventral longitudinal trunk in light brown, spiracles in grey, wing tracheae in light green, leg tracheae in dark green, visceral tracheae in dark blue, commissures in dark brown, additional tracheae in light blue. Abbreviations: ahlt, anterior hind leg trachea; amlt, anterior midleg trachea; dat, dorsoanterior trachea; dc, dorsal commissure; dlt, dorsal longitudinal trunk; dpt, dorsal prothoracic trachea; dt, dorsoposterior trachea; flt, frontleg trachea; fwt, front wing trachea; hwt, hindwing trachea; lc, leg commissure; lpt, lateral prothoracic trachea; phlt, posterior hind leg trachea; pmlt, posterior midleg trachea; ts, thoracic spiracles; vc, ventral commissure; vlt, ventral longitudinal trunk; vpt, ventral prothoracic trachea; vt, visceral trachea. Scales bars: one mm.

The ventral (vlt in Fig. 2, colored in light brown) and dorsal (dlt in Fig. 2, colored in purple) longitudinal trunks are well developed and their general structure is similar to the adult’s system (described in detail below). All main trachea described in the imago are also present in larva, but they are less developed, i.e., they have less branches and are usually shorter.

The head is supported by the dorsal and ventral longitudinal trunks, from which numerus trachea branch. These tracheae also resemble the trachea of the imago (described in detail below).

Commissures of the dorsal (dc in Fig. 2, shown in dark brown) and ventral (vc in Fig. 2, shown in dark brown) longitudinal trunk are found in the head as brain commissure, in the mesothorax (two of each trunk) and in all following segments (one of each trunk).

The leg tracheae are shown in dark green in Fig. 2. The foreleg trachea (flt in Fig. 2) originates from the mesothoracic spiracle. The mesothoracic and hind leg trachea are similar to the imago supplied by their adjacent spiracles and comprise an anterior middle leg trachea (amlt), a posterior middle leg trachea (pmlt), an anterior hind leg trachea (ahlt) and a posterior hind leg trachea (phlt).

The wing tracheae are shown in light green in Fig. 2. They are well developed in the meso- (forewing trachea, fwt) and metathorax (hindwing trachea, hwt) but lack distal wing trachea. The hind wing trachea that runs between the metathoracic (ts2) and the first abdominal spiracle (as1) is particularly well developed.

The visceral tracheae are shown in dark blue in Fig. 2. They supply the digestive system and the inner organs. Visceral tracheae in the head and thorax are small and derive from the ventral longitudinal trunk. The visceral tracheal in all abdominal segments derive from the ventral longitudinal trunk or the spiracles and continue from there anteriorly. Visceral tracheae are especially in abdomen not so well developed as in the imago. They are strait and do not form loops.

In the prothorax, three additional tracheae (dpt, lpt, vpt in Fig. 2 shown in light blue) that do belong to any of the previous described types originate from the mesothoracic spiracle and supply the lateral, dorsal and posterior wall of prothorax.

The tracheal system of the larva of T. molitor is shown in Fig. 1B. It is described in detail in Raś, Iwan & Kamiński (2018).

The pupa

The tracheal system of the pupa of G. tibialis is shown in Figs. 1C & 2B.

The composition of the pupal tracheal system is almost identical to the above described larval one. Differences are found in the form and development of the distal wing trachea of the forewing. In contrast to the larva, it reaches into to the freshly developed elytra. In their shape and morphology, the remaining tracheae undergo a gradual change towards the condition of the imaginal tracheal system.

The tracheal system of the pupa of T. molitor is shown in Fig. 1D. It is described in detail in Raś, Iwan & Kamiński (2018).

The imago

The tracheal system of the imago of Gonopus tibialis is shown in Figs. 1E & 2C. Two major tracheae, the longitudinal trunks run longitudinally through the entire length of the animal and connect each spiracle with the ones anteriorly and posteriorly. They are the ventral longitudinal trunk and the dorsal longitudinal trunk. The ventral longitudinal trunk (vlt; light brown in Fig. 2) supports the ventral parts of each segment. Within the head capsule, several branching tracheae originate from it: the ventral mandibular muscles trachea, the ventral brain trachea that supplies the brain, the fronto-clypeal trachea, the maxillary trachea, the mandibular trachea and the optico-antennal trachea that penetrates the central part of the compound eyes and antennae. In the thoracic and abdominal segments, the visceral tracheae (vt) derive from the ventral longitudinal trunks and run towards the middle of the body toward the alimentary canal. The ventral longitudinal trunk of both body sides are connected by various commissures (vc, dark brown in Fig. 2). Within the head capsule, the ventral head commissure from which the pair of the labial trachea originates. The ventral prothoracic commissure connects both left and right ventral longitudinal trunks in the prothorax. Two pairs of tracheae originate from this commissure: the prothoracic ganglion tracheae that run posteriorly and suboesophageal ganglion tracheae that continue anteriorly. The mesothoracic ganglion and the extrinsic muscles of the meso- and metathoracic coxae are also penetrated by the tracheae of the ventral meso- and metathoracic commissures. Ventral commissures also occur in each abdominal segment with a spiracle.

The dorsal longitudinal trunk (dlt; red in Fig. 2) supplies the dorsal part of each segment including the notae. Within the head, it is responsible for delivering breathing gases to the brain, the posteriodorsal part of the mandibular abductor, the optic nerve, parts of the compound eyes and antennae. In the abdominal segments, it gives rise to the dorsoanterior trachea (dat) that originate in the anterior part of the segment and the dorsoposterior trachea (dt) in the posterior part near the respective spiracles. The dorsal longitudinal trunk has the following commissures (dc, in dark brown): the brain commissure in the head capsule which gives rise to the brain trachea. The relatively thin dorsal commissures occur in all post-cephalic segments. There are two of them between the meso and the metathoracic spiracle and one between the dorsal longitudinal trunks of all following spiracles. These commissures sometimes were compressed and invisible on models of the tracheal system.

The legs are supported by the leg tracheae (dark green in in Fig. 2): the prothoracic leg is supported by the foreleg trachea (flt) that runs anteriorly from the mesothoracic spiracle. One branch of this trachea also runs into the anterolateral prothorax. The leg commissure (lc) connects the leg trachea from each body half just before they enter the coxae. The mesothoracic and metathoracic legs are both supplied by trachea coming from the spiracles anteriorly and posteriorly of the respective leg. In case of the mesothoracic leg they are the anterior midleg trachea (amlt; from the mesothoracic spiracle) and the posterior midleg trachea (pmlt; from the metathoracic spiracle). For the metathoracic leg they are the anterior hind leg trachea (ahlt; from the metathoracic spiracle) and the posterior hind leg trachea (phlt; from the first abdominal spiracle).

The wing trachea (light green in Fig. 2): The mesothorax has a well-developed forewing trachea (fwt) that forms an arch between the meso- and the metathoracic spiracle. Six distal wing tracheae (dwt) branch from it and run into the elytron. The anterior three of those branches (that form the lateral trachea within the elytron) branch from the forewing trachea while the posterior three (the mesal elytral trachea) derive from a common stem from the wing trachea. Five of these trachea run within the 1., 3., 5., 7. and 9. elytral intervals while the last one is in the epipleuron. The hindwing trachea (hwt) is much more slender than the forewing one. It connects the metathoracic spiracle with the first abdominal one. It lacks any distal wing tracheae.

The visceral tracheae (vt) are dark blue in Fig. 2. They branch from spiracles or the ventral longitudinal trunks and supply the alimentary canal and other internal organs. The visceral tracheae of the head and thorax are small and derive from the ventral longitudinal trunk. In the abdomen, G. tibialis has very well developed and long visceral trachea that support not only the abdomen but also reach anteriorly into the pterothorax. They form distinctive loops.

Next to these serial tracheae that occur in several or all segments, there are some additional tracheae (light blue in Fig. 2) that originate from single spiracles and cannot be assigned to any of the previous categories. Within the prothorax there are three massive prothoracic tracheae (light blue in Fig. 2), the ventral prothorax trachea (vpt), lateral prothorax trachea (lpt) and dorsal prothorax trachea (dpt); the trachea direct towards the anterolateral, the anterior and the anterodorsal part of prothorax, respectively.

In general, abdominal tracheae are characterized by distinct branches which comprise a main branch with a large diameter and numerous smaller ones originated from this main branches (e.g., the dorsal anterior tracheae dap, the dorsal posterior trachea or the ventral commissures). Most abdominal tracheae concentrate near the digestive and reproductive systems.

The tracheal system of the imago of T. molitor is shown in Fig. 2F. It is described in detail in Raś, Iwan & Kamiński (2018).

Morphometrics of the tracheal system

The volume of the studied tracheal system of G. tibialis is 2.46 mm3 (±0.87; n = 9) in the larva, 3.94 mm3 (±0.95; n = 2) in the pupa and 16.38 mm3 (±5.18; n = 9) in the imago. The respective volume in comparison to the body volume is shown in Fig. 3A. It covers 0.884%, 0.631% and 2.239% of the entire body volume in the three life stages. 19% (2.96 ± 0.87 mm3) of the studied tracheae of the imago of G. tibialis belong to the cephalo-prothoracic section (green in Fig. 1), 36% (5.82 ± 2.20 mm3) to the pterothoracic (red in Fig. 1) and 45% (9.01 ±  2.87 mm3) to the abdominal section (blue in Fig. 1) (Fig. 3B). Detailed values for every specimen of T. molitor are found in the supplementary material (Raś, 2020, S.2). Figure 3C shows the tracheal system growth rate for the ratio between body tissue volume (bv) to tracheal system (tv) volume for the two transitions during the metamorphosis (larva/pupa and pupa/imago). In the case of the transformation from larva to pupa, the growth slope of the fitted line is 0.69 (intercept = −2.986) indicating negative allometry, and the square of the correlation is R2 = 0.794 (p = 0.0002). In contrast, the growth rate of the pupa/imago transition shows positive allometry (3.03; intercept = −17.333), but the value of the square of the correlation is low (R2 = 0.303, p = 0.0789). The values for T. molitor are taken from Raś, Iwan & Kamiński (2018) and the volume and other measures calculated are shown in the respective figures with those of G. tibialis. The diameter of the trachea (longitudinal trunks, leg und wing trachea) that run posteriorly from the mesothoracic spiracle are shown in Fig. 3D.

Figure 3 Morphometric measures of the tracheal system of Gonopus tibialis and Tenebrio molitor.

(A) Tracheal volume in relation to the body volume for the larva, pupa and imago of Gonopus tibialis and Tenebrio molitor (absolute values can be found in the supplementary material (Raś, 2020). (B) Relative size (in %) of the tracheal system of the three different sections of the imagos of Gonopus tibialis and Tenebrio molitor. CPS, cephalic-prothoracic section; PTS, pterothoracic section; ABS, abdominal section. (C) growth ratio of the tracheal systems in relation to body volume for the two major developmental transitions larva/pupa (straight line) and pupa/adult (broken line) for Gonopus tibialis and Tenebrio molitor. (D) Diameter of the tracheae connecting the mesothoracic and metathoracic spiracles of Gonopus tibialis and Tenebrio molitor. Image of the respective tracheae on the left, diagram with the measured values on the right. Colors indicate the respective tracheae. Abbreviations: amlt, anterior midleg tracheae; dlt, dorsal longitudinal trunk; pmlt, posterior midleg trachea; vlt, ventral longitudinal trunk; wt, wing trachea.

The results of the Gaussian mixture models show the density distribution of the reconstructed tracheal system (Fig. 4A). The analyses were performed for 3 adult specimens of both studied species. Detailed visualization and measures are in the supplementary material (Raś, 2020, S.3)

Figure 4 Morphometric measures of the tracheal system of Gonopus tibialis and Tenebrio molitor.

(A) density plot obtained by Gaussian mixture model representing tracheal system distribution within the body of Gonopus tibialis and Tenebrio molitor in dorsal and lateral view. The warmer the color, the denser the tracheae in this area. (B) Mean relative spiracle volume in relation to the complete body volume of the pterothoracic section (PTS with the thoracic spiracles ts) and the abdominal section (AS with the abdominal spiracles as) for Gonopus tibialis and Tenebrio molitor. Absolute values can be found in the supplementary material (Raś, 2020). (C) Principal component analysis for the spiracle distribution in different developmental stages (larva, pupa & imago) of Gonopus tibialis and Tenebrio molitor.

Spiracle volumes

The volumes of the spiracles of the adults of G. tibialis range between 0.009 ± 0.006 and 0.219 ± 0.082 mm3 (n = 8) (all values provided in 25 S.2). The mesothoracic one has by far the largest volume (0.22 ± 0.08 mm3) and is 4.1 times more voluminous than the metathoracic one (0.05 ± 0.02 mm3). The volumes of abdominal spiracles range between 0.009 ± 0.006 mm3 (8th abdominal spiracle) and 0.038 ± 0.013 mm3 (1st abdominal spiracle). These results are confirmed by pairwise comparisons using the Wilcoxon rank sum test (p = 0.01) where ts1 and ts2 are significantly bigger than all other spiracles. The values of T. molitor are taken from Raś, Iwan & Kamiński (2018). Fig. 4B compares the size of the spiracles for both species in relation to their body volume.

The 3D data on the spiracle distribution for the different developmental stages (larva, pupa, imago, Figs. 1A–1C) were analyzed with principal components analyses (PCA). The results for the two most informative components are shown in Fig. 4C. The most part of the transformation (84%) is covered by principal component 1, which represents spiracle changes in the coronal plane during the development (in sequence of transitions larva/pupa/imago). The postembryonic development of G. tibialis is accompanied by significant differences in all developmental stages of this species (ANOVA, p < 0.001, F = 35.45). Our analyses retrieve significant interspecific differences between the imagines of both studied species (ANOVA, n = 9, p = 0.001, F = 57.7) but not between the larvae (ANOVA, n = 9, p = 0.755, F = 0.569).

Thin-plate spline deformation grids were used to show the intraspecific differences between the transitions between the developmental stages (larva to pupa and pupa to adult) for both species (Fig. 5A) and for a interspecific comparison between the stages of the two species (Fig. 5B). In G. tibialis, an extreme widening and rounding of the abdomen occurs between the pupal and the imago stage which is not observed in T. molitor (Fig. 5A). At the same time the distance between the pterothoracic and the first abdominal spiracle shortens (Fig. 5A). The comparison of the three life stages between the two species (Fig. 5B) shows, that there is almost no difference in habitus between the larvae. In the pupa the distances between the mesothoracic, metathoracic and 1st abdominal spiracle starts to alter between the species. The major differences are observed between the imagos and include mainly the extreme widening of the abdomen and the length of the metathorax.

Figure 5 Comparative spiracle distribution illustrated with thin-plane spline grids.

Red dots indicate the thoracic spiracles, blue ones the abdominal spiracles. (A) Intraspecific comparison between different developmental stages (larva/pupa & pupa/imago) within Tenebrio molitor (on the left side) and Gonopus tibialis (on the right side). (B) Interspecific comparison between Tenebrio molitor and Gonopus tibialis for the larva, pupa and imago.

Body volume

The volume of the entire body excluding the tracheal system for the two species is shown in Fig. 6A. The results of the Kruskal Wallace test analyses show the significant differences of the volumes of all developmental stages of G. tibialis (p < 0.001, chi-squared = 14.3, df = 2), and also between the imagos (p < 0.001, chi-squared = 12.8, df = 1) but not larvae (p = 0.054, chi-squared = 3.7, df = 1) of both studied species.

Figure 6 Logarithmic volume of the complete body, the subelytral cavity and the alimentary canal and 3D model of the tracheal system including the subelytral cavity.

(A) Logarithmic volume of the complete body, the subelytral cavity and the alimentary canal for Gonopus tibialis and Tenebrio molitor (absolute values can be found in the supplementary material (Raś, 2020) (B) 3D model of the tracheal system including the subelytral cavity (in transparent blue) of Gonopus tibialis in dorsal view. Spiracle positions (ts = thoracic spiracles; as = abdominal spiracles) indicated by circles. (C) Sagittal (on the left) and longitudinal (on the right) sections through the µ-Ct scans of the body of Gonopus tibialis illustrating the subelytral cavity in strongly expanded (above) and reduced (below) state and the openings of spiracle into it. Abbreviations: el, elytra; sc, subelytral cavity; sp, spiracle; tr, trachea.

Subelytral cavity (Fig. 6)

The subelytral cavity of G. tibialis is the space between the dorsal body wall and the firmly fused elytra (Figs. 6B, 6C). A video of it is provided in supplementary material (Raś, 2020). All spiracles except the mesothoracic ones open into this cavity (Fig. 6B). In G. tibialis it is sealed towards the outside while in T. molitor, it can be opened in order to expand the wings. In G. tibialis, the volume of the subelytral cavity varies strongly and ranges between 23.18 and 432.7 mm3 (Figs. 6A; 6C shows the extreme cases) while in T. molitor it ranges between 6.9 and 33.7 mm3, which represents 3.7–41.3% and 4.4–21.1% of the total body volume (n = 8 for T. molitor and n = 9 for G. tibialis). The mean volume of the subelytral cavity is 21.4 ± 10.0 for T. molitor and 137.2 ± 131.9 mm3 for G. tibialis. All data is provided in the supplementary material (Raś, 2020, S.2, S.4).

Figure 7 3D reconstructions based on µ-Ct scans of the digestive system of Tenebrio molitor (A & C) and Gonopus tibialis (B & D) in dorsal view, Outline of the beetles shown in grey.

(A) & (B) show only the digestive system while (C) & (D) also show the tracheal system and its interaction with the digestive system. Tracheal system separated into the three defined sections: cephalic-prothoracic (CPS, in green), pterothoracic (PTS, in yellow) and abdominal (ABS, in blue). Abbreviations: an, anus, lo, loops in the abdominal digestive system, ph, pharynx.

Digestive system (Fig. 7)

The massively developed digestive system of G. tibialis (Figs. 7B, 7D) is characterized by two massive loops in the posterior pterothoracic and abdominal area (lo in Fig. 7). The one of T. molitor (Figs. 7A, 7C) is much more delicate in comparison with the one of G. tibialis. It lacks those massive loops and has only a small protrusion. The volume of the alimentary canal for G. tibialis varied between 27.0 and 46.0 mm3 (mean value: 36.5 ± 9.1; n = 5), and for T. molitor 1.0−1.8mm3 (1.3 ± 0.4; n = 3), which are 3.1−5.2% and 0.6−1.2% of the total body volume respectively. All data is provided in the supplementary material (Raś, 2020, S.2, S.4).

Discussion

Tracheal system and flight

Our results show a clear correlation between the ability to fly and the dimensions and shape of the adult tracheal system in the pterothorax (the segments that hold the wings) in the two studied species. They thus confirm our hypotheses 1 that we see differences in this area between the winged and the unwinged species. In beetles, this specifically affects the metathorax that holds the hindwings which are reduced in the wingless species. The front wings are sclerotized protective plates (elytra) that do not contribute to creating uplift during flight and are still retained in wingless species. The metathorax of the wingless G. tibialis is much shorter compared to total body length than the one of the winged T. molitor (Raś, Iwan & Kamiński, 2018) (10.5 ± 0.5% of the total body length vs 17.0 ± 1.1%; n = 10 respectively; see details in supplementary material (Raś, 2020, S.2). This shortening, which is also seen in the thin split grids (Fig. 5B) is associated with the massive decrease of the flight musculature, which is commonly observed in wingless insects across all insect groups (e.g., Wipfler et al., 2015; Fabian, Schneeberg & Beutel, 2016). As these flight muscles are extremely demanding in terms of energy supply due to their enlarged and specialized mitochondria (Edwards & Ruska, 1955; Wigglesworth & Lee, 1982), their reduction strongly reduced the demand for oxygen and thus the size of the tracheal system. This is clearly shown by several of our performed analyses: the Gaussian mixture models point out that the tracheal density is highest in the pterothorax of T. molitor but not in the apterous G. tibialis. In the winged T. molitor 54% (1.38 ± 0.30 mm3 n = 9) of all studied tracheae are found in the pterothorax while it is only 36% (5.82 ± 2.20 mm3, n = 8) in the wingless G. tibialis. On an anatomical level, adult G. tibialis lost the metathoracic distal wing tracheae that supports the hindwings in the studied flying beetle, although their precursor trachea is still present in their larvae. Another strong indication for the different oxygen demand in the pterothorax between the two species is the relative volume of the metathoracic and the first abdominal spiracles that support the metathoracic flight musculature. In relation to whole body volume, they are in T. molitor 3.9 times (metathorax) and 9.6 times (1st abdominal segment) larger than in G. tibialis (Fig. 4B). In summary, our data clearly show various differences between the studied winged and the wingless species in the pterothoracic tracheal system and their association with flight. As the ability to fly is the ancestral condition for beetles (McKenna et al., 2019), we conclude that the modifications in G. tibialis represent a derived condition. We only studied two species which allows only limited conclusions about general trends and thus the generalization of the stated hypothesis (Garland Jr & Adolph, 1994). As no other study ever compared the tracheal volume or any other of the studied parameters between winged and unwinged species, we cannot directly compare our data with those of previous research. However, we can derive indirect conclusions from known morphological adaptations of wingless insects. Various other insects including beetles that reduced the wings show similar morphological adaptations as the ones we observed here including shortening of the respective thoracic segments, simplification of the respective tergal plates and reduction of musculature (e.g., Wipfler et al., 2015). We thus hypothesize that at least some of the observed correlations between flight muscle reduction and the resulting changes in shape and morphometrics of the tracheal system also occur in other flightless beetles or apterous insects from other groups. Of course, this has to be tested by future research with a broader taxon sampling.

Special adaptations in the tracheal system of G. tibialis

In the following part of the discussion, we show that the tracheal system of the wingless desert beetle G. tibialis comprises several modifications that would either render the wings inoperable, result in a heavy abdomen or reduce the O2 supply of the pterothoracic segments and are thus incompatible with a flying insect. These modifications all stand in relation with the arid environment that G. tibialis lives in. They thus support our 2nd hypothesis that the reduction of the thoracic tracheal system which supplies the flight musculature apparently created the possibilities to evolve new specializations in this species which are incompatible with a functional flight apparatus. We also discuss to which extend these modifications are also found in other flightless desert beetles and assess the question to which extend the found observation in the studied species can be transferred to these other species.

The abdomen of adult G. tibialis is strongly broadened in contrast to the one of T. molitor as shown in our thin split grids (Fig. 6B). This creates space for the enlarged digestive system of G. tibialis which is according to our data approximately five times larger in relation to the whole body volume than the one of T. molitor. Numerous abdominal loops are formed by the mid- and hindgut in imagos of this species while adult T. molitor as well as the larvae of both species only have a single regular loop (Fig. 7). One of those digestive loops of G. tibialis is located in the meso- and metathorax, where flight musculature is found in T. molitor. Thus the strongly developed digestive system of G. tibialis is incompatible with the musculature required for active flight. As a consequence, the tracheal system of the abdominal section which mainly supports the digestive system (Fig. 1) is also enlarged in G. tibialis. It comprises 45% of the entire studied tracheal system while this value is only 30% in T. molitor (Wilcox test, W = 71, p-value <0.001). Our Gaussian mixture models also clearly show that the point with the densest tracheal system is found in the abdomen in G. tibialis but not in T. molitor (Fig. 4A). A possible explanation for this enlarged digestive system and the associated tracheae of G. tibialis is the arid environment it inhabits. The digestive system is also responsible for water resorption (Grimstone, Mullinger & Ramsay, 1968) which is next to water retention a driving force in desert animals (Addo-Bediako, Chown & Gaston, 2001). It is likely that the strong enlargement of the digestive tract is involved in this process. Additionally, a longer digestive tract can absorb more nutrients from the food which is important when low nutritive food such as sandy substrate of arid and semiarid area is consumed.

Another speculated function of the enlarged abdomen is to provide space for storage of water or energy in the abdominal fat bodies (Arrese & Soulages, 2010). Adults of G. tibialis are adapted to survive the adverse period between rainy seasons where little to no food is available. The same applies to the periodic access to water (Zachariassen, 1996). However, most desert beetles including G. tibialis are strongly sclerotized in order to reduce water loss via the cuticle which makes the required increase in volume difficult or impossible. It was hypothesized that another modification of the tracheal system in wingless desert beetles, the subelytral cavity is associated with the solution to this problem (Draney, 1993). This cavity comprises the space between the tightly fused elytra and the soft dorsal surface of the abdomen and is thus hermetically sealed air chamber connected to the tracheal system inside the beetle. It thus allows the expansion of the abdomen on the expanse of this cavity. We addressed this hypothesis for the first time with measurements of the volume of the subelytral cavity in several individuals and we can show that the volume of the subelytral cavity in the studied specimens varies strongly between 10 to 185 mm3 in the studied specimens of G. tibialis. We thus provide strong evidence for the idea that the subelytral cavity allows the expansion of fat bodies and other storage organs in the abdomen.

Our results also show that the presence of the subelytral cavity has other far reaching consequences for the tracheal system of G. tibialis. With the exception of the mesothoracic one, all spiracles of this species open into this chamber. This is also observed in other beetles which have such a cavity. As the subelytral cavity has little to no interchange with the outside air but rather forms a sealed chamber, it is generally considered to be part of the tracheal system (Duncan, 2003). As the spiracles that open into this cavity cannot participate in air exchange with the outside, this task has to be completely performed by the mesothoracic spiracles (e.g., Kuusik et al., 2016). Consequently, these only openings of the tracheal system are extremely enlarged. According to our data, it is 4.2 times more voluminous than the second largest one (the mesothoracic one) in G. tibialis while in the winged T. molitor the meso- and metathoracic ones are distinctly more voluminous (Fig. 1F; Raś, 2020, S.2). Similar enlarged mesothoracic spiracles were also observed in other flightless desert beetles (Zachariassen, 1991; Draney, 1993; Duncan & Byrne, 2000), which provides evidence that the observations we made for the tracheal system of G. tibialis might also apply to other wingless desert beetles with a subelytral cavity. We would have also expected that the diameter of the longitudinal trunk tracheae that run posteriorly from the mesothoracic spiracle is enlarged in the wingless species in order to ensure a sound support of the entire posterior body. However, our data shows that their relative diameter compared to the body volume is much smaller than those of the winged species (Fig. 3D). The same applies to the wing trachea. Apparently, the musculature of an active flight apparatus in the metathorax requires more oxygen than the supply of the entire body from a single spiracle. However, the fact that the metathoracic spiracle does supply oxygen from the outside affects the composition of the trachea of the mesothoracic leg. The anterior one that comes from the mesothoracic spiracle is comparatively much larger in G. tibialis than in T. molitor while for the posterior one it is vice versa. Comparable data about the diameter of the trachea connecting the mesothoracic spiracle with the subelytral cavity has not been gathered for any other species but the studied ones. We thus have no data to compare it to other wingless desert beetles with similar enlarged spiracles which would allow to transfer the observations for these two species to a general statement about wingless beetles.

It was shown that the air inside the subelytral cavity exhibits extremely high humidity (Zachariassen, 1991; Draney, 1993; Duncan, 2002). When artificially opened, the beetles experience a drastically increased water loss ranging between 60% and 70% (Zachariassen, 1991; Kuusik et al., 2016). It thus might function –next to the above described space for the abdomen expansion –as a storage organ for humidity collected in the respiratory system. Additionally, it can also participate in cooling the animal as some desert beetles ventilate it when exposed to higher temperatures (Bolwig, 1957; Hadley, 1970). The stored humidity is thus released and serves as an effective way of evaporative cooling.

As outlined above, the subelytral cavity, i.e., the transformation of the space below the elytra into a part of the tracheal system has several advantages for the studied desert beetle including the storage or energy and water in the abdomen, the retention of humidity and the ability to cool the animal. At the same time, it requires several modifications which are incompatible with an operational flight apparatus. They include the need to firmly fuse the elytra which would make the expansion of the hindwings impossible, the air exchange with a single pair of spiracles (that would not supply enough oxygen for the flight musculature) and large and heavy storage organs in the abdomen. The same applies to the enlarged digestive system that fills most of the abdomen and would be too heavy for a flying species. Although we only studied a single species with these modifications, we have good reason to assume that the above provided arguments also apply to other beetles with subelytral cavities as they show similar morphological adaptations: the elytra are firmly sealed in all species with a subelytral cavity (in fact, this is part of the definition of a subelytral cavity), they only have a single pair of spiracles that open to the exterior and show similar enlarged abdomen. We thus assume that the tradeoff between the ability to fly and the various benefits of the subelytral cavity applies to all species with this structure and thus speculate that our 2nd hypothesis is generally valid with respect to the subelytral cavity. However, this of course has to be tested by future research with a broader taxon sampling. As a subelytral cavity is present in various not-related flightless groups of beetles, it must have evolved several times independently which renders the general validity of our hypothesis even more interesting and worthy to study. We also cannot address the question if these modifications are the cause for the loss of wings (Gallegoa, Verdúa & Lobob, 2018 and references therein) or a result of it.

Tracheal system and development

The third hypothesis of the present manuscript deals with the question, when the observed modifications of the tracheal system of the two studied species develop. Our observations clearly confirm the hypothesis that this happens during the pupa.

The spiracle position data which can be considered as a proxy for the outer body shape shows that the larvae of both species show a very similar spiracle distribution and thus habitus. The distance between the spiracles of all segments is almost identical. The same applies to the left and right spiracles of one segment which implies that all segments have similar widths. Consequently, the larvae of both species cluster in an identical morphospace in our principal component analysis of the spiracle positions (Fig. 4C) and our thin plate spline grids also show almost no differences between the two studied species (Fig. 5A). The differences in the adult spiracle position, i.e., the extreme widening of the abdomen in G. tibialis and the increase of space between the metathoracic and 1st abdominal stigma in T. molitor develop during pupation (Fig. 5A). The studied pupae are already clearly distinguishable based on the first principle component of our analysis of the body shape which mostly represents the widening and shortening of the body (Fig. 4C). The adults are even better distinguishable.

The relative size of the tracheal system volume compared to the complete body volume is higher in the larva of G. tibialis than in the one of T. molitor (0.884 vs. 0.623), which indicates an increased level of demand for gas exchange. This might be correlated with the biology of this species as its soil larva digs corridors which requires well-developed fore legs and muscles (Schulze, 1962; M Raś & D Iwan, pers. obs., 2013). In the pupal stage where the flight apparatus is developed, this relation is reversed and the tracheal system of T. molitor becomes comparatively bigger which is also retained in the imagos (2.239 in G. tibialis versus 2.756 in T. molitor). In both species, the ratio between tracheal system volume and body tissue volume is significantly higher in imagos than in larvae but in T. molitor it increased by the factor 4.4 while in G. tibialis it is only 2.5. This stands most likely in correlation with the flight ability and the associated need for oxygen for the wing muscles of T. molitor. Previous studies also suggested that the relative size of the tracheal system is correlated with the body size of the animal (Kaiser et al., 2007). As the two studied species differ in their size, this might also account for some differences.

The strong differences in the developmental stages between the two species are also shown in the ratio between the volumes of the tracheal system and the whole body for the transitions between the different life stages (i.e., the change from larva to pupa and from pupa to imago). In the case of G. tibialis, there is a continuous increase of this ratio. This implies that both the tracheal and the body volume grow in both transitions, although in different speeds which results in negative allometry in the larva/pupa transition (i.e., the tracheal system grows slightly lower than the body; slope: 0.69, intercept = −2.986) and positive allometry in the pupa/imago transition (the tracheal system grows stronger than the body; 3.03; −17.333). In T. molitor, the tracheal system volume and the body tissues volume grow at the same rate in the larva/pupa transition (isometry; 1.23, −2.717). During the transition between pupa and adult however, the volume of the body is reduced while that of the tracheal system strongly grows (negative allometry, 2.18, 5.059) (Fig. 1E), which is most likely correlated with the massive flight musculature. As we do not have comparable data for any other insect, we cannot address if similar patterns are also overserved in other species.

It was previously shown that almost all parts of the insect’s body including the flight apparatus development, the musculature and the reproductive system experience significant reorganization during the pupal stage (Belles, 2011; Ge et al., 2015; Hall & Martín-Vega, 2019). Our results show that this also applies to the tracheal system and that the pupa is the key evolutionary stage that mitigates between the uniform tracheal systems of the larvae and the strongly variable adults with their different adaptations.

Conclusions

The present study provides the first direct comparison between the tracheal systems of a winged and a wingless insect. We can show that wing reduction has severe consequences for the tracheal system of the darkling beetle G. tibialis. It results in strong reductions of the relative tracheal volume in the pterothoracic segments which hold the flight apparatus in winged species. Instead this species shows several adaptations towards its arid habitat such as a subelytral cavity which leads to respiration via a single pair of spiracles, a shortened metathorax or an extremely enlarged abdominal digestive system that also covers parts of the metathorax where flight musculature is located in the winged species. Our results show that these modifications would be incompatible with a functional flight apparatus but we cannot answer the question whether they are the cause or a result of the loss of flight. Our developmental results show that all of these modifications form during the pupal stage while the larvae of both species are extremely similar and thus underline the extreme evolutionary importance of the metamorphosis.

The authors thank Juliane Vehof (Bonn) for help with figures. BW and MR thank David Hasselhoff for emotional support.

Additional Information and Declarations

Competing Interests

Author Contributions

Field Study Permissions

Data Availability

The authors declare there are no competing interests.

Marcin Raś conceived and designed the experiments, performed the experiments, analyzed the data, prepared figures and/or tables, authored or reviewed drafts of the article, and approved the final draft.

Benjamin Wipfler conceived and designed the experiments, prepared figures and/or tables, authored or reviewed drafts of the article, and approved the final draft.

Tim Dannenfeld analyzed the data, prepared figures and/or tables, authored or reviewed drafts of the article, and approved the final draft.

Dariusz Iwan conceived and designed the experiments, performed the experiments, authored or reviewed drafts of the article, and approved the final draft.

The following information was supplied relating to field study approvals (i.e., approving body and any reference numbers):

Collection of material was approved by the Ministry of Environment and Tourism of Namibia (permit number: 1738/2012).

The following information was supplied regarding data availability:

Raś M. 2020. Postembryonic development of the tracheal system of beetles in the context of aptery supplementary material. Harvard Dataverse, V2. https://doi.org/10.7910/DVN/P1HRUQ

The raw data is available at MorphoSource:

000385386, https://doi.org/10.17602/M2/M385386;

000386223, https://doi.org/10.17602/M2/M386223;

000386229, https://doi.org/10.17602/M2/M386229;

000386234, https://doi.org/10.17602/M2/M386234;

000386239, https://doi.org/10.17602/M2/M386239;

000386244, https://doi.org/10.17602/M2/M386244;

000386249, https://doi.org/10.17602/M2/M386249;

000386254, https://doi.org/10.17602/M2/M386254;

000386267, https://doi.org/10.17602/M2/M386267;

000386435, https://doi.org/10.17602/M2/M386435;

000387853, https://doi.org/10.17602/M2/M387853;

000387858, https://doi.org/10.17602/M2/M387858;

000387863, https://doi.org/10.17602/M2/M387863;

000387868, https://doi.org/10.17602/M2/M387868;

000387873, https://doi.org/10.17602/M2/M387873;

000387878, https://doi.org/10.17602/M2/M387878;

000387883, https://doi.org/10.17602/M2/M387883;

000387888, https://doi.org/10.17602/M2/M387888;

000388023, https://doi.org/10.17602/M2/M388023;

000388028, https://doi.org/10.17602/M2/M388028

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
