# Peer review of "Postembryonic development of the tracheal system of beetles in the context of aptery and adaptations towards an arid environment"

_PeerJ, doi:10.7717/peerj.13378_

## Round 0.1 · original submission · Major Revisions

Dear Dr. Raś and colleagues:


Thanks for submitting your manuscript to PeerJ. I have now received two independent reviews of your work, and as you will see, one reviewer recommended rejection. The other reviewer was only a bit more enthusiastic, raising some serious concerns. I agree with the concerns of both reviewers.

Fortunately, there is a lot of criticism here for you to consider. If you choose to address these concerns, I encourage you to revise your work and resubmit. However, I strongly encourage you to take into account all of the concerns raised by both reviewers.

Please note that issues related to methodological repeatability (missing information) and limitations with using two taxa to infer adaptation/evolution need to be seriously handled in your revision.

Thanks again for submitting your work to PeerJ.

Good luck with your revision,

-joe

·

Basic reporting

This study focuses on the description of post-embryonic development of adult trachea in the wingless beetle G. tibialis. The authors employed a 3D morphometric analysis to analyze and compare the tracheal morphology of wingless beetle G. tibialis and the winged beetle T. molitor. The results reveals a reduction of tracheal density in the thoracic segments as well as relevant differences in spiracle size. Interestingly, such reduction of the thoracic tracheal in G. tibialis correlates with the reduction of the flight muscles and the consequent development of the elytra cavity. In addition, those tracheal differences are achieved during pupal development as tracheal larva of both species are almost identical. Finally, the authors associate the diminish of tracheal system in G. tibialis with its specific habitat adaptions to arid environments, as it correlates with an increase of the digestive system and the arise of the elytra cavity, two features related to water retention.
In general, the study contributes to the understanding of tracheal development and its possible association to adaptation of different habitats, being of interest for a broad number of readers. That said, I think that the quality of the manuscript can be improved by clarifying and elaborating on a few points, which I detail below. It would particularly be useful to include more information on the morphological description of the G. tibialis trachea, the T. molitor larval tracheal network and on the subelytra cavity. I also suggest a substantial rewriting of the results and discussion sections to make it more appealing to a general audience. These changes would significantly increase the repeatability and clarity of the study.

Experimental design

1. My main concern of the study is that the Ms is not well structured. The results are not well organized and therefore are difficult to follow. Some figures are cited in the text in a wrong order (Fig 1F is mentioned after Fig 2) and some results are not illustrated in the figures (Subelytrala cavity and digestive system plots are not included in any figure). In addition, some results are only well described in the discussion, whereas others are only found in this section. Thus, I strongly recommend restructuring the results and the discussion to make the Ms more readable. Below I give some advises to improve the report.
2. Fig. 1 illustrates the tracheal morphology of G. tibialis in imago (A), pupa (B) and larva stage (C). As development proceeds from larva to imago, it should be better to show them in the opposite order, from the larva to the adult. In addition, it would be useful to indicate the position of the spiracles in each stage. Finally, including the 3D reconstruction of T. molitor tracheal tree of larva, pupa and adult in this figure would help the reader to understand the difference and the posterior analysis presented in panels D and E.
3. Description of tracheal of G. tibialis (line 152-161) are difficult to follow as some of the tracheal branches are not well depicted in the 3D reconstruction. For example, the following sentence “The pupa of G. tibialis differs from T. molitor by having only rudimentary basal hind wing trachea” is not well illustrated in the figures, making difficult to interpret the results.
4. The statistical analysis should be integrated in the description of the results as it is a tool but not a result by itself. Therefore, it should not have the category of a subtitle.
5. Spiracle distribution. In this part, Fig 3B is again mentioned before Fig 3A. The description of the relative spiracle size in both species, should be presented in this part (plot of Fig1F). Thus, I recommend to move panel F of Fig 1 to Fig 3. I do not understand what it is represented in Fig3B. According to Fig 3C, spiracle distribution in panel B of G. tibialis represents the imago and the pupa, in other words the transition from pupa (lower diagram) to imago (upper diagram). The draws of stages in the middle, therefore are confusing and should be changed. In addition, a larval diagram of T. molitor spiracles, should be included to better understand the spiracles distribution in all transitions.
6. Interestingly, the thin plate spline grid shows clearly a progressive reduction of as1 from larva to imago in G. tibialis whereas in T. molitor as1 is clearly expanded, this could be related to the reduction of the muscle wing mass. Surprisingly, this result is only mentioned in the discussion. Regarding to that, is it this expansion already observed in T. molitor larva or it is again a feature developed during metamorphosis?
7. Body volume, Subelytral cavity and Digestive system analysis are not illustrated in any figure. The plots showing the significative differences should be added in the Figures and described properly in the results. In addition, an 3D dimensional image of Digestive system (without the tracheal system) of the two species should be shown as it would help to visualize the difference on size and the loops formation.
8. Subelytral cavity 3D dimensional image connected to the bigger spiracle is required.
9. Line 333. The relative size of the larva tracheal tree should be reported in the results with an appropriate plot in the Figures.
10. Line 345-354. This paragraph should be in the results and not in the discussion.
11. This study compares the tracheal system of two different species. However, it would be interesting to discuss the different morphology in species with winged and wingless individuals like the ants. In fact, a recent study has analyzed the morphological differences between the queens (winged) and the workers (wingless) and their implications in ecological adaptations
12. In conclusions line 371, impossible should be replace to incompatible.

Validity of the findings

In summary the data presented in this report is robust and the conclusions are supported by the results. However a deep restructuration of the results and discussion sections is required to significantly increase the repeatability and clarity of the study.

Reviewer 2 ·

Basic reporting

Overall the document is well written. There are some places where the language should be checked for clarity. The authors missed some relevant references in the introduction Kaiser et al. (2007). Increase in tracheal investment with beetle size supports hypothesis of oxygen limitation on insect gigantism. Proceedings of the National Academy of Sciences USA 104:13198-13203.; Greenlee et al. (2013) Hypoxia-induced compression in the tracheal system in the caterpillar, Manduca sexta L. Journal of Experimental Biology. 216:2293-2301. PMID: 23531813; Helm et al. (2018) Characterizing metamorphic development in the solitary bee Megachile rotundata using micro-computed tomography. Arthropod Structure and Development. https://doi.org/10.1016/j.asd.2018.05.001.

Experimental design

The authors used 3D morphometric analysis of the tracheal system to examine differences between one species of wingless beetle and one winged beetle. The authors show beautiful images of 3D tracheal reconstructions, and they used a large sample size for this type of study.

Validity of the findings

The conclusions are overstated. The use of only two species to draw conclusions about evolutionary adaptations is incorrect (Garland, T., Jr., and S. C. Adolph. "Why not to do two-species comparative studies: limitations on inferring adaptation." Physiological
Zoology 67.4 (1994): 797-828). This is especially problematic when one of the species has a very different lifestyle, living in the desert, and one is considered a cosmopolitan warehouse pest. Thus, the first hypothesis cannot be supported as a general statement on winged versus wingless species. The second hypothesis also cannot be supported because the authors only looked at one species that is desert adapted. The third hypothesis is clearly supported with the images.

Additional comments

Line Comment
45 This suggests they are controlling their environment. Consider rephrasing.
54 Please correct trachea to tracheae, which is the plural form.
73 Unclear what is altering trachea arrangement.
117-141 This part of the methods section is formatted with lots of very short paragraphs or singular sentences.
255-256 The statement that the digestive system required reduction of flight musculature is making the assumption that the evolution of the increased digestive system drove the reduction of flight musculature. However, it could easily have happened the other way. This is where a truly comparative study that looked at multiple species across a known phylogeny may be able to answer such a question.
Fig. 1 Please arrange the developmental stages in order A) larva, B) pupa, and C) imago. What life stage was used to compute the data in Fig. 1D? In Fig 1F, suggest to use transparent symbols for the two species, because there is so much overlap. Possibly consider using a log scale for the y-axis so that we can see what the spiracle volumes are.
Fig. 3 It is unclear what is being shown in 3B and 3C. It is not clear where the comparison between the two species is on 3C.

---

## Round 0.2 · accepted · Accept

Dear Dr. Raś and colleagues:

Thanks for revising your manuscript based on the concerns raised by the reviewers. I now believe that your manuscript is suitable for publication. Congratulations! I look forward to seeing this work in print, and I anticipate it being an important resource for groups studying beetle anatomy and physiology. Thanks again for choosing PeerJ to publish such important work.

Best,

-joe

·

Basic reporting

The authors have sufficiently responded to all of my concerns on the structure of the report.

Experimental design

The authors have sufficiently responded to all of my experimental issues.

Validity of the findings

The Ms meets all the standars of the journal.